# Precise language responses versus easy rating scales—Comparing respondents' views with clinicians' belief of the respondent's views

**Sverker Sikström** \*, Alfred Pålsson Höök, Oscar Kjell

Lund University, Lund, Sweden

\* sverker.sikstrom@psy.lu.se

## Abstract

### Background

Closed-ended rating scales are the most used response format for researchers and clinicians to quantify mental states, whereas in natural contexts people communicate with natural language. The reason for using such scales is that they are typically argued to be more precise in measuring mental constructs; however, the respondents' views as to what best communicates mental states are frequently ignored, which is important for making them comply with assessment.

### Methods

We assessed respondents' ($N = 304$) degree of depression using rating scales, descriptive words, selected words, and free text responses and probed the respondents for their preferences concerning the response formats across twelve dimensions related to the precision of communicating their mental states and the ease of responding. This was compared with the clinicians' ($N = 40$) belief of the respondent's view.

### Results

Respondents found free text to be more precise (e.g., *precision d'* = .88, *elaboration d'* = 2.0) than rating scales, whereas rating scales were rated as easier to respond to (e.g., *easier d'* = −.67, *faster d'* = −1.13). Respondents preferred the free text responses to a greater degree than rating scales compared to clinicians' belief of the respondents' views.

### Conclusions

These findings support previous studies concluding that future assessment of mental health can be aided by computational methods based on text data. Participants prefer an open response format as it allows them to elaborate, be precise, etc., with respect to their mental health issues, although rating scales are viewed as faster and easier.

**Data Availability Statement:** All relevant data for this study are publicly available from the OSF repository (https://osf.io/47bdz/).

**Funding:** Marianne och Marcus Wallenbergs Stiftelse (MMW-2021.0058). "AI-based language models for improving diagnostics, monitoring, and outcome of depression and anxiety". Vinnova. Förbättrad diagnostisering av mental hälsa med beskrivande ord och artificiell intelligens. (2018-02007). Kamprad Foundation. Förbättrad diagnostik för psykisk ohälsa hos äldre: implementering av beslutsstöd baserat på beskrivande ord och artificiell intelligens, ref # 20180281 The funders had no role in study design, data collection and analysis, decision to publish, or preparation of the manuscript.

**Competing interests:** Sverker Sikström and Oscar Kjell are shareholders of WordDiagnostic AB. This does not alter our adherence to PLOS ONE policies on sharing data and materials.

# Introduction

Language is the natural way of communicating mental states among most groups in our society. The notable exceptions are behavioural scientists, where close-ended rating scales are the dominant method for examining respondents' states of mind. For example, in social psychology articles, there are typically 20 scales in each article, where 87% of them can be categorised as rating scales [1]. The most likely reason for this is that rating scales are easy to collect and quantify and have been found to have reasonably high validity with regard to the construct being measured.

The argument that rating scales should be preferred over open-ended responses due to high validity has recently been challenged by interdisciplinary studies based on open-ended questions and progress in natural language processing (NLP). Question-based computational language assessments (QCLA), where respondents are asked one or more open-ended questions and the responses are analysed with NLP and machine learning methods, has become a promising alternative to rating scales. For example, Kjell et al. [2] showed that descriptive word responses concerning harmony in life, satisfaction with life, worry, and depression can be quantified so that they correlate well with corresponding, well-established rating scales. Furthermore, although similar constructs, such as depression and anxiety, tend to correlate strongly with rating scales, they are better differentiated by QCLA [2]. In addition, pictures portraying facial expressions are more accurately differentiated when described by participants using descriptive words rather than rating scales [2].

These findings are particularly interesting as opened-ended questions evaluated by humans have not been shown to increase the validity of mental health assessment. Friborg and Rosenvinge [3], for example, allowed students and assistants to evaluate the responses from open-ended questions concerning mental health and found that their evaluations did not increase the statistical prediction of mental health one year later in relation to closed-ended rating scales. Similarly, data on the degree of drinking collected with telephone interviews showed that closed-ended responses provided different results compared to human evaluations of opened-ended responses [4]. Hence, the possibility that NLP methods can make a contribution, in addition to human performance, needs to be investigated.

Opportunities to use QCLA have increased as a result of recent progress in computational methods. For example, BERT (Bidirectional Encoder Representations from Transformers; [5]) is a language model based on deep learning neural networks that use transformers [6], which has greatly improved the accuracy of NLP models. Recently, BERT was used to predict rating scales with an accuracy ($r$ = .85) that challenges the theoretical upper limits of test-retest reliability [7]. This was accomplished using multiple free texts and descriptive word responses regarding harmony in life and satisfaction with life as inputs. Further, Sikström et al. (in progress) showed that a combination of word responses and rating scales diminishes the number of missed diagnoses (keeping false alarms constant) of depression and anxiety by approximately half compared to state-of-the-art rating scales (the Patient Health Questionnaire, PHQ-9 [8], and the Generalized Anxiety Disorder Scale, GAD-7 [9] when validated by criteria related to DSM-5 [10]. This suggests that the validity of the PHQ-9 or the GAD-7 may be improved by adding open-ended questions assessed with computational language assessment. Freely generated word responses analysed with NLP have been found to correlate with the symptoms in DSM-5 that are associated with Major Depressive Disorder (MDD) and Generalized Anxiety Disorder (GAD; [11]. Finally, QCLA allows for the description of mental constructs on the group level by, for example, visualising in word clouds what words are associated with high or low degrees of depression [2]. This also extends to the individual level where words such as *separated*, *alone*, and *unemployed* provide a better understanding of who

is a potential patient than what is communicated with rating scales. In summary, these studies suggest that QCLA may have a considerably high accuracy and thus the potential to rival the measurement validity of rating scales. However, few studies have compared the ecological validity of language responses compared to rating scales.

We argue that the computational assessment of language responses to health-related questions is not typically done in clinical settings, and that this method can improve the validity of assessment. While interviews are part of current clinical assessment methods (e.g., [12]), current practice excludes computational methods. Nevertheless, previous studies provide evidence that the computational assessment of language data can improve the validity of the assessment beyond that of rating scales, for example, with a better categorisation of facial expressions [2] and improved correlations with cooperative behaviour [13]. Therefore, these data suggest that computational assessment of language data may potentially improve the validity of assessment of MDD and GAD in research and clinical settings. For instance, computational assessment of text data and rating scales correlates substantially higher with the MINI interview for GAD and MDD compared to rating scales alone [14]; however, further research is needed to support this claim. Little data is available on direct comparisons between how well the computational assessment of text data on health questionnaires compares with experienced clinical evaluations of the same data. Beyond the question of validity, the computational assessment of text data can also provide other advantages as it can be used when there is no clinic available to conduct a clinical interview or provide a second opinion to such interviews. Moreover, it supplies richer data than rating scales alone.

To study the respondents' views of the assessment is important, as a working relation between patients and clinician over time is essential for motivating patients to reveal sensitive information.

In this study, we wanted to compare the preferences of a population of respondents taking mental health tests (i.e., patients or non-patients that may seek assessment) with those of clinicians (who select and administrate the assessments) with respect to meta-beliefs concerning the respondents' preferences. We were particularly interested in whether the clinicians' beliefs about the respondents' preferences matched the real preferences as rated by the respondents. This is an important question as the clinicians should acknowledge respondents' preferences when selecting the assessment instruments to establish efficient communication.

Given the current trend in research that considers language a promising and accurate measure of mental states, we were primarily interested in how respondents perceive language-based response formats, including free texts, descriptive words, and selected words from a pre-defined list. We investigated whether respondents with or without diagnoses, and clinicians' view of these language-based response formats, were more *precise* in communicating their mental states and whether rating scales are perceived as easier to respond to.

Given the recent progress in the accuracy of computational methods in analysing text responses, and that language is the natural way for people to communicate their mental states, we hypothesise (H1) that respondents view free text as more *precise* in communicating their mental states compared to rating scales. Nevertheless, we also hypothesise (H2) that rating scales are *easier* to respond to than free text responses. Furthermore, we hypothesise (H3) that formats related to keywords have ratings in between the free recall and rating scales formats. Finally, because language is the natural way for people to communicate their mental state, we hypothesise (H4) that respondents are more likely to prefer free text responses while communicating with clinicians regardless of what clinicians believe the respondents prefer.

We tested these hypotheses by first allowing respondents answer questions related to depression. The choice of depression, rather than other mental disorders, was governed by the fact that it is a common mental disorder, and that our research group has successfully focused

on measuring depression using NLP in previous studies (e.g., [2]). We used four response formats–free text, descriptive words, selected words, and rating scales–where the order of the listed formats went from very open to closed. In the analyses, we focused primarily on comparing the free text and rating scales, where we expected the descriptive and selected word condition to have intermediate rating scores. Once again, the choice of these four formats was based on the fact that we have previously used them in earlier studies (e.g., [2, 7]). We then asked the respondents to rate their view of the response format concerning the following 12 dimensions. How precise it was to communicate their mental states with the response format was measured by eight questions about *precision* in communication, the *nature* of their mental states, evoking or *reinstating* emotions, how they could *elaborate* their answer, how they could *think* more, how the format provides opportunities to consider the construct in *different* ways, if it is a *natural* way to communicate mental states, and how well they can *relate* to communicating with the format. How easy it is to respond to the response format was measured by three questions about how *easy*, *fast*, and *demanding* it was to use. Finally, one item measured the *preference* for communicating using the response format. This study was not preregistered.

## Materials and methods

### Participants

Were recruited from the Prolific website (https://www.prolific.co/). The inclusion criteria were US nationality with English as a first language and an age of 18 years or older.

The clinician condition included participants who stated they were employed as a nurse, psychologist, or medical doctor. There were 92 participants on Prolific that met the criteria for taking part in the clinician condition, where 62 of these participants started the study and 40 completed it, 22 dropped out prior to completing the study. This group was chosen because we wanted to target professionals that work with patients on a daily basis.

In the respondents' condition ($N$ = 304), a pre-screening was conducted to identify a population where approximately half of the respondents self-reported that they had an ongoing major depression diagnosis (MDD) or generalized anxiety disorder (GAD). The other half were respondents who did not report these diagnoses, hereafter referred to as the *control* group. We chose to include participants with MDD or GAD as these are two major mental health disorders with high comorbidity (i.e., rating scales of measuring them correlates highly). Participants that matched the pre-screened condition were invited to participate until the predefined number of participants was reached. The data collection lasted for approximately 17 hours on Prolific. Data of participants that did not complete the study was not saved. Participants with MDD and GAD have been systematically investigating using QCLA in several previous studies (e.g., [2, 7, 11]). A drawback with this choice was that a smaller group of the participants only had a GAD diagnosis (N = 21), whereas most of them had MDD, which is the focus of the current study (N = 145 and sometimes also GAD).

The mean age was 32.7 years, ranging from 18 to 74 years, with a standard deviation of 11.5 years. There were 75 men and 224 women, and 1 preferred not to say. To participate in the study, all were given the incentive of receiving 1.88 pounds.

### Procedure

The study consisted of a survey (constructed in LimeSurvey) that was shared with potential respondents via Prolific. Participants were first asked for informed consent to participate in the study and instructed that they could leave at any time. This was followed by instructions for how to answer the survey (see the S1 Appendix). Participation was anonymous. The study was divided into two phases, where Phase 1 was related to the assessment of mental health in

different response formats, and Phase 2 involved answering questions related to their perception of the response format.

In Phase 1, participants in the respondent condition were asked to answer questions regarding whether they had been depressed during the last two weeks. They were assessed four times, each time with a different response format, namely free text, descriptive words, selected words, and rating scales. The order of the response formats was randomised and there was no time between the tests. Participants in the clinician condition were asked to read the questions so that they understood the response format but were not to respond to the questions. The reason why clinicians should not respond to the questions was to simulate their natural working environment as non-respondents.

Phase 2 collected the 12 preference ratings towards the four response formats used in Phase 1. The participants were asked for their own preference, whereas the clinicians were asked to estimate the participants' preference. The order of the statements was randomised, and the survey took approximately 15 minutes to complete.

## Material

In the free text format, respondents were asked to write a descriptive text with a minimum of 20 words and a maximum number of 1000 words. In the descriptive word format, respondents were asked to type five descriptive words [2]. In the select words format, the instructions were to choose five words from a set of 30 words, where the selection of words was taken from the most commonly-used words in previous studies that used the descriptive word format [2]. Finally, in the rating scales format respondents used a traditional Likert-scale [8] consisting of a set of statements to which the respondents were asked to respond by choosing one of five responses from 0 (Not at all) to 4 (Nearly every day).

Preferences towards the four response formats were assessed using 12 statements (see the S1 Appendix) created by the authors with the purpose of measuring the precision in communicating mental health, the cognitive ease of response, and the preference for using the formats. The statements related to precision were as follows: allowing one to *elaborate*, how *precisely* could you communicate, *think* through depression, *reinstate* emotions, communicate the *nature* of depression, thinking about depression in a *different* way, and how well you *relate*. Statements related to cognitive ease included *easy*, *demanding*, and *fast* to respond. Finally, there was the one statement that participants would *prefer* to use. Here we chose Likert scales as the response format because this is currently the most accepted way to quantify preferences and because the QCLA has not been validated for these preferences. There were five alternatives– 0 for "Don´t agree at all", 1 for "Don´t agree to some extent", 2 "Neither agree nor disagree, 3 "Agree to some extent", and 4 "Fully agree". The scale for the *demanding* question was reversed to make it compatible with responses to the *easy* and *fast* questions, and it is therefore henceforth referred to as *less demanding*.

## Data analysis

The data was divided into the clinician condition and the respondent condition, and then further divided into the four response categories (rating scales, selected words, freely generated words, and free text) and 12 rating scales related to the preference of the response categories. Descriptive analysis was then conducted by reporting the mean responses of each rating scale for each response category. Following this, inferential statistics were calculated using *t*-tests, with a focus on comparing the preferences related to the rating scales with those associated with the free text responses. We used a Bonferroni correction for multiple testing of the rating scales ($N = 12$), and the alpha level was set to .05.

## Ethics

The study was evaluated by the Etikprövningsnämden (https://etikprovningsmyndigheten.se/) in Sweden. The applicant requested an advisory opinion for a very similar study (i.e., the same population, design and questions related to mental health) and the Ethics Review Authority maintained it had no objection to the research described in the application (EPN Dnr 2020–00730). Thus, the study is in accordance with the Swedish Ethical Review Act (SFS 2003:460). Participants were, however, required to give written informed consent prior to joining the study. All participants were adults (i.e., above 18 years), and the consent forms were stored in the response data file. All data were collected anonymously.

## Results

### The respondent condition

**PHQ-9 severity scores.** The PHQ-9 scores (ranging from 0 to 27) were calculated for the respondents and classified into none-minimal ($N = 52$, range 0–4), mild ($N = 57$, range 5–9), moderate ($N = 67$, range 10–14), moderately severe ($N = 69$, range 15–19), and severe ($N = 59$, range 20–27).

The results are summarised in Table 1A. Fig 1 shows the respondent condition using the rating scales as the baseline. Free text and rating scale responses were compared with two-tailed paired t-tests.

Compared to the rating scales, the respondents viewed the free text as more precise than the rating scales (Hypothesis 1) for 6 of the 9 conditions: more *precise* ($t(302) = 8.16$, $p < .0001$, $d' = .88$); better in *reinstating* ($t(302) = 6.28$, $p < .0001$, $d' = .62$) their emotions and expressing the *nature* ($t(302) = 5.31$, $p < .0001$, $d' = .54$) of their mental state; *elaboration* ($t(302) = 20.87$, $p < .0001$, $d' = 2.0$); deeper *thinking* ($t(302) = 6.20$, $p < .0001$, $d' = .62$); and made them think in a *different* way ($t(302) = 2.63$, $p < .0001$, $d' = .26$) However, three measures showed no significant ($p > .05$) difference between the free text and rating scales: how *naturally* respondents could *relate* to answering the questions in the response format and to what extent they would *prefer* to respond in a format for a clinician.

Furthermore, the respondents rated the rating scales as easier than the free text response formats (Hypothesis 2) in the three measures aimed at capturing this, that is, the rating scales were perceived as *easier* ($t(302) = –7.62$, $p < .0001$, $d' = –.67$), *faster* ($t(302) = –11.70$, $p < .0001$, $d' = –1.13$), and less *demanding* ($t(302) = –9.95$, $p < .0001$, $d' = 1.03$). The objective time to complete the rating scale ($m = 60.9$ s) was also significantly faster than the time to complete the free text response ($m = 186.3$ s, $t(302) = 10.0$, $p < .0001$, $d' = .58$). The selected word condition ($m = 50.7$, $t(302) = –3.41$, $p < .0001$, $d' = -.20$), however, was somewhat faster than the rating scales, whereas the descriptive words condition was slightly slower ($m = 77.3$, $t(302) = 3.52$, $p < .0001$, $d' = .20$).

Consistent with Hypothesis 3, the descriptive word and selected word formats displayed results that were nominally in between the free text and the rating scales in all conditions, with a significant difference between the rating scales and free text responses.

**Clinician condition.** The results for the clinicians are summarised in Table 1B. These were consistent with the results found in the respondents' conditions in all measures, except for the 'thinking about it in a *different* way' ratings, where there was no significant difference between free text and rating scale condition.

All significant findings in the respondents' conditions, following correction for multiple comparisons ($N = 12$), had $p$-values < .001, and for non-significant $p > .05$ alpha was set to .05.

**Table 1. Means and standard deviations of preferences for respondents and clinicians' estimation of the respondents' preferences.**

a) Respondent condition

| Questions | Rating Scale | Free Text | Descriptive Words | Selected Words | *p* |
|---|---|---|---|---|---|
| Elaborate | 0.98 (1.14) | 3.27 (1.02) | 1.64 (1.10) | 1.44 (1.20) | < .0001 |
| Precise | 2.11 (1.23) | 3.05 (1.00) | 2.29 (1.12) | 2.23 (1.18) | < .0001 |
| Think | 1.66 (1.28) | 2.45 (1.35) | 1.79 (1.27) | 1.79 (1.27) | < .0001 |
| Reinstate | 1.35 (1.17) | 2.10 (1.34) | 1.59 (1.21) | 1.50 (1.20) | < .0001 |
| Nature | 2.37 (1.10) | 3.03 (1.07) | 2.38 (1.05) | 2.29 (1.09) | < .0001 |
| Different | 1.09 (1.08) | 1.39 (1.20) | 1.35 (1.16) | 1.39 (1.19) | < .0001 |
| Natural | 2.40 (1.19) | 2.53 (1.35) | 2.04 (1.21) | 2.15 (1.24) | .2060 |
| Relate | 2.69 (1.04) | 2.81 (1.13) | 2.45 (1.00) | 2.46 (1.00) | .1460 |
| Prefer | 2.44 (1.00) | 2.52 (1.13) | 2.16 (0.96) | 2.15 (0.97) | .4450 |
| Easy | 3.07 (1.00) | 2.21 (1.28) | 2.15 (1.17) | 2.30 (1.20) | < .0001 |
| Less demanding | 2.92 (1.04) | 1.71 (1.3) | 1.99 (1.17) | 2.12 (1.18) | < .0001 |
| Fast | 3.06 (1.04) | 1.81 (1.24) | 2.02 (1.09) | 2.23 (1.10) | < .0001 |

b) Clinician condition

| Questions | Rating Scale | Free Text | Descriptive word | Selected Words | *p* |
|---|---|---|---|---|---|
| Elaborate | 1.05 (1.17) | 3.21 (0.91) | 2.09 (1.02) | 1.67 (1.25) | < .0001 |
| Precise | 1.93 (1.12) | 2.86 (1.08) | 2.51 (1.08) | 2.56 (1.26) | < .0001 |
| Think | 2.02 (1.14) | 2.77 (1.32) | 2.28 (1.12) | 2.07 (1.10) | 0.0006 |
| Reinstate | 1.70 (1.01) | 2.47 (1.39) | 2.07 (1.12) | 2.30 (1.08) | 0.0044 |
| Nature | 2.53 (0.96) | 2.67 (1.17) | 2.81 (0.85) | 2.67 (0.87) | 0.0018 |
| Different | 1.70 (1.17) | 1.79 (1.26) | 1.60 (1.09) | 1.88 (1.20) | 0.5334 |
| Natural | 2.35 (1.21) | 2.33 (1.29) | 2.58 (0.98) | 2.47 (1.12) | 0.6518 |
| Relate | 2.47 (1.01) | 2.28 (1.10) | 2.63 (0.82) | 2.70 (0.89) | 0.9311 |
| Prefer | 2.84 (0.87) | 1.44 (1.16) | 2.28 (1.01) | 2.47 (1.10) | 0.4302 |
| Easy | 2.95 (1.07) | 1.56 (1.31) | 2.21 (1.08) | 2.47 (1.24) | < .0001 |
| Less demanding | 2.66 (1.08) | 0.79 (0.99) | 1.63 (1.00) | 2.12 (1.07) | < .0001 |
| Fast | 2.77 (1.00) | 1.00 (1.09) | 1.95 (0.87) | 2.26 (1.05) | < .0001 |

*Note*. The table shows mean values with standard deviations in parentheses. The *p*-values show whether rating scales differed significantly from the free text values using two-tailed paired t-tests. The rows are ordered after their ratings in the free text minus the rating scale condition in the respondent condition.

## Comparing respondents' and clinicians' conditions

Respondents had significantly stronger preferences (Hypothesis 4, see Fig 2A) for free text responses over rating scales compared to clinicians' estimate of their *preference* ($t(345) = 5.25$, $p < .0001$, $d' = 0.85$) following correction for multiple comparisons (N = 12), as well as for the following four preferences without correction: *less demanding* ($t(345) = 2.59$, $p = .0099$, $d' = 0.42$), *easy* ($t(345) = 2.11$, $p = .0353$, $d' = 0.34$), communicate the *nature* of their mental state ($t(345) = 2.22$, $p = .0269$, $d' = 0.36$), and *fast* ($t(345) = 1.97$, $p = .0491$, $d' = 0.32$).

## Comparing patients with controls

The control group rated elaboration in free text responses as providing a greater advantage in ratings scales compared to patients with self-reported ongoing MDD or GAD ($t(329) = 3.51$, $p = .0005$, $d' = 0.39$), whereas patients reported faster response rating scales as providing the larger advantage compared to free text responses ($t(329) = -3.79$, $p = .0002$, $d' = -0.42$) following correction for multiple comparisons (see Fig 2B).

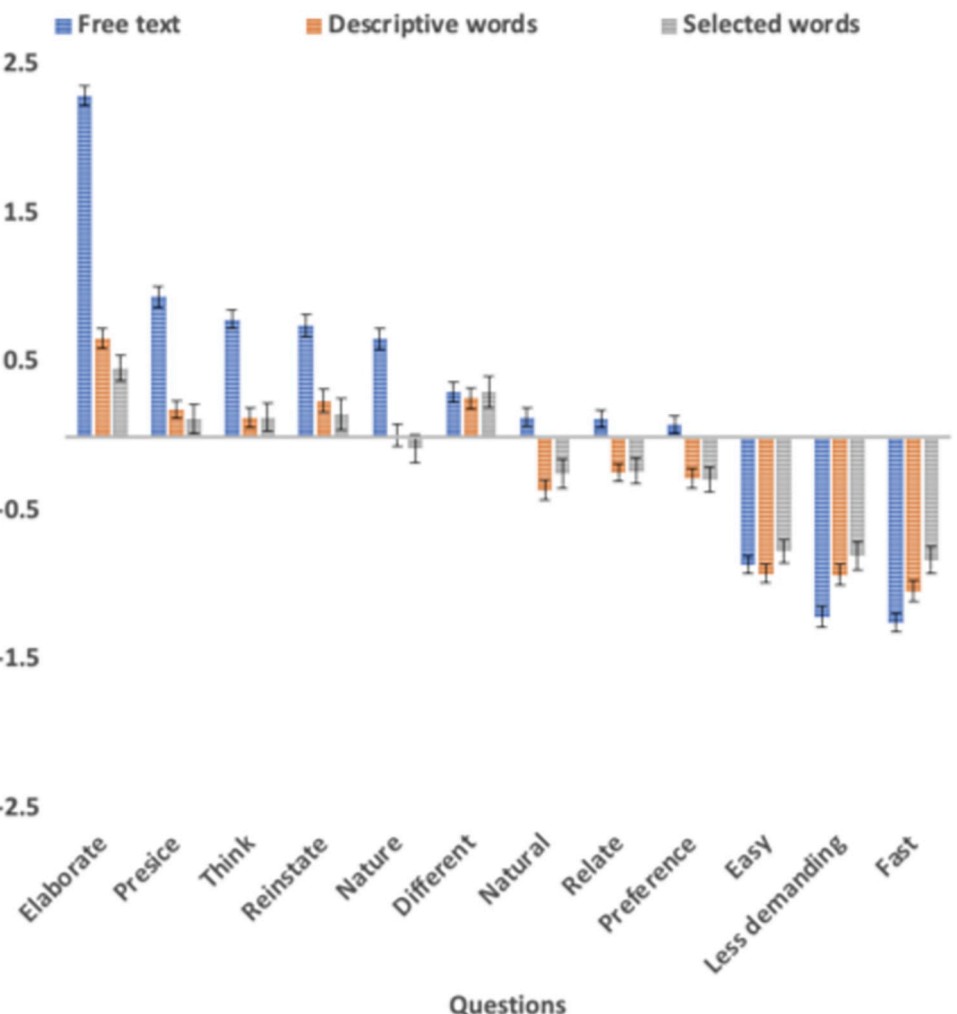

**Fig 1. Free text, descriptive words, and selected word responses using rating scales as the baseline for the respondent condition.** *Note*. The y-axis shows the mean response and standard errors for free text (blue), descriptive words (red), and selected words (grey) minus the mean response for rating scales. The x-axis is ordered by the free text minus rating scales value.

## Discussion

The results clearly show that respondents and clinicians viewed the free text response format as more precise for communicating their state of mind compared to rating scales. This finding is consistent with recent work showing that language-based responses, analysed by state-of-the-art computational language models, show high accuracy and are competitive with and sometimes outperform rating scales [2, 8, 13, 14].

Although our data do not allow causal claims, we speculate that one reason why text responses may be perceived as more accurate is that they allow for more specific *elaborations* of the respondents' state-of-mind, making them think more about the construct. In addition, our data show that the respondents rated free text responses as enabling them to view the construct in a *different* way, and previous research suggests that this may be an important aspect for improved mental health [15]. The results did not, however, show any difference between the free text and rating scales in the dimensions of being *natural*, easier to *relate* to, or more

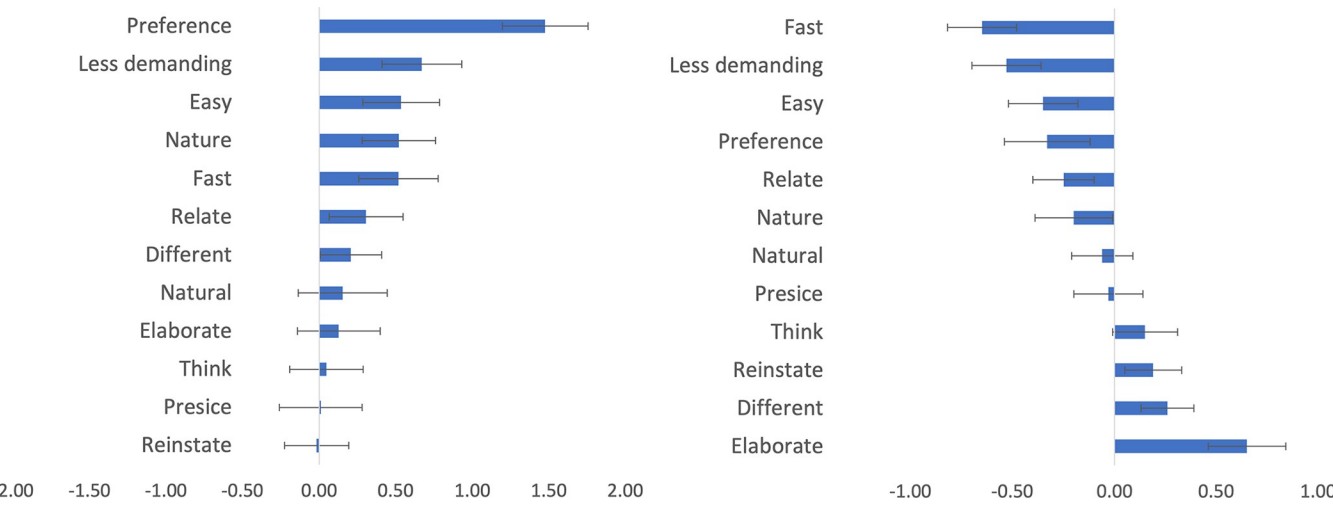

**Fig 2.** A. Respondents' preference for free text responses over rating scales using clinicians' estimates of the respondents' preference as the baseline. B. The control group's preference for free text responses over rating scales using diagnosis as the baseline. *Note*. The figure shows the mean ratings of free recall responses minus the mean ratings for rating scales. Fig 2A shows respondents' ratings minus clinicians' ratings where a positive value indicates that respondents' preference for free text responses over rating scales is greater than clinicians' estimate of their preference. Fig 2B shows the control group ratings minus the ratings of participants with diagnoses of depression or anxiety, where positive values indicate that the control group preferred free text response over the rating scales of those with a diagnosis. The error bars are standard errors.

*preferred*. A possible reason for this is that *natural* and *relate* are concepts that can be interpreted in different ways, and open formats are less constrained and therefore harder to connect to these constructs.

The results also showed that both respondents' and clinicians' estimates (of the respondents' ratings) considered rating scales *faster*, *easier*, and *less demanding* than free text responses. Moreover, the rating scales also took less time to perform than free text responses. The fact that rating scales are easier is obviously an advantage; however, this may be contrasted to the finding that they are rated as less precise. A possible interpretation of this is a trade-off between speed and accuracy, where the rated precision of free recall responses can be exchanged for a slower response [16]. Furthermore, the selected word responses were actually answered faster than the rating scale, whereas the descriptive words were answered just slightly slower than the rating scales. Currently, rating scales are the most common response format in Prolific [1], but the speed of response to open-language questions may increase as the respondents become more accustomed to these response formats.

Respondents had the same *preference* for rating scales when communicating their mental states to a clinician; Nevertheless, the respondents also indicated a stronger preference for free text formats compared to the clinicians estimate of the respondents' ratings. Interestingly, this was the rating where the difference between respondent and clinician ratings was the largest ($d' = 0.85$). Furthermore, clinicians exaggerated how slow, demanding, and hard (as measured by *fast*, *less demanding*, and *easy*) the respondents rated free recall responses. Thus, on several questions, clinicians could not estimate how respondents rated the questions related to the response formats on depression. The findings cannot easily be explained with the possibility that the clinicians have little knowledge of the respondent's point of view, as they could also have simulated the respondents answer with their own preferences. An alternative explanation is that the clinicians' preferences are different from those of the respondents.

Another difference is that the respondents had the opportunity to answer the four response formats related to depression, which the clinicians were instructed not to do. This design was

intentionally setup to simulate a situation where it felt natural for respondents to answer questions, but not clinicians. We argue that this setup is more interesting to study as it is closer to the real-life interaction between respondents and clinicians, compared to the alternative design where clinicians and respondents would have been treated identically.

Free text response is a more open format compared to freely generated descriptive words or selected words. We therefore expected that the preferences for the descriptive word responses would show results that were in between the free text and rating scales, something also found in the majority of the questions.

The findings that participants rate free text response to be more precise in communicating mental health compared to rating scales is consistent with recent findings showing that QCLA has a high validity in measuring mental health. For example, Kjell et al. [2, 7] used natural language processing algorithms (e.g., Latent Semantic Analysis [LSA]; [17]; BERT, [5]) to map the text responses to vectors representing participants' descriptions of their mental health. They then used machine learning (ML, multiple linear regression) to build a model to predict ratings scales for mental health (e.g., PHQ-9 and GAD-7). The results revealed that text responses can be accurately used to predict rating scales. An advantage with this method is that it provides an accurate and standardised procedure to assess mental health in a format that feels natural for participants (patients) to communicate their mental health. A disadvantage, however, is that participants and clinicians need to understand and trust the results. These computational findings suggest that text data, consistent with participants' ratings, can be used to make quantitative assessments of mental health with high accuracy.

Rating scales have been a standard method to measure mental states in psychological research for the last century. Several standard methods for taking advantage of, understanding, and applying rating scales are currently in use. Examples of such methods are Receiver Operating Curves (ROC, [18]), Signal Detection Theory (SDT), and Item Response Theory (IRT, [19]). Computational language assessment allows us to map freely generated texts into a single quantifiable dimension associated with, for example, a rating scale. This can be achieved by developing a machine learning method that processes a set of texts and generates a scale that correlates well with an associated rating scale (e.g., see [2, 7]). In addition, the generated unidimensional semantic scale can then be applied to the methods developed for rating scales (e.g., ROC, SDT, IRT). Further research is, however, needed to more carefully analyse possibilities and limitations while applying rating scale methods to semantic scales.

Writing about mental health issues is not only a method of assessment; it also influences the severity of mental health. A large body of empirical evidence shows that repeated writing about mental health improves the severity of mental health issues such as anxiety and depression [20], whereas the improvement that follows mental health rating scales is modest. Pennebaker and Beall [21] originally suggested the expressive writing paradigm, where four episodes of 15 minute writing sessions about a traumatic episode diminished the severity of posttraumatic stress disorder (PTSD). However, re-exposure to traumatic events should also be handled with care. Jon Frederickson [20] argued that "getting information without regulating the fragile patient's anxiety would be harmful, not healing" and reminded clinicians that re-exposure interventions should be combined with strategies of manage distress related to traumatic events ([20] p. 205). However, a recent meta review of the expressive writing paradigm suggests that the improvement in mental health related to, for example, depression, is a stable and reproducible phenomenon [21].

The proposed method of question-based computational language assessment can be founded on general mental health questions (i.e., describe your mental health) or more specific queries related to symptoms (describe your appetite, concentration, etc.). General questions have the advantage of not priming, or activating, a participant to focus on aspects that may not

be relevant for a particular patient, whereas specific questions may help participants focus on aspects that they otherwise would have missed. Thus, general, and specific questions may have complementary purposes, where general questions should proceed specific questions. Previous studies [7] have shown that the computational assessment of all open-ended questions related to specific MDD symptoms correlates with standard rating scales of depression (i.e., PHQ-9); however, cognitive and emotional aspects yielded higher predictability than secondary criteria of behavioural aspects. Furthermore, combining several open-ended questions improves accuracy compared with single questions [7, 14]. More importantly, as computational language assessment is based on machine learning, it may pick up relevant cues that are implicitly related to the to-be-measured construct. For example, the excessive usage of the first pronouns in freely generated texts (i.e., "I") is associated with depression, although the text may not necessarily be related to mental health [22]. Furthermore, a recent study on a clinical sample of MDD and GAD demonstrates that combining rating scales with open-ended questions provides more accurate assessments compared to rating scales alone [14].

## Limitations

We do not claim that all assessment of mental health is well supported by computational language assessment. While this article focuses on depression, prior work from our lab has also successfully included studies on anxiety, harmony in life, and satisfaction with life (e.g., [2]). Common to these assessments is that they relate to the emotional and/or cognitive states/experiences of the participants, where such states and experiences are commonly communicated in language. However, in our view, language based on assessment methods may be less suitable for mental health issues related to cognitive performance, for example, ADHD or dementia. Furthermore, how suitable computational language assessment methods are for psychotic disorders or mental health issues related to anxiety or depression, for example, obsessive compulsive disorder (OCD), is a topic for future research.

Furthermore, cultural factors may influence assessment of depression, and this may be particularly relevant for computational language assessments as different populations may use different languages. Although this topic has not been systematically investigated, studies of computational language assessment in both young and old individuals may provide hints, where the training of data on older populations generalises well to young ones and vice versa; however, the former training was significantly more successful than later (Sikström & Kelmendi., in progress).

Another concern is that machine learning that is used by QCLA may be influenced by biases. For example, women are more likely to experience anxiety, leading to the concern that the machine learning may incorrectly use female language as evidence for anxiety. This effect has been labelled the Alignment Problem [23]. Depending on the training data, this problem may influence various ethnical groups, and may be a disadvantage that may be particularly problematic for non-White Anglo-Saxon Protestants (WASP).

Although the study showed clearly significant results, the number of participants classified as clinicians was rather limited, despite the fact that we invited all clinicians available on Prolific given our inclusion/exclusion criteria. The limited size of participants did not allow us to control for various demographic factors such as age, gender, education, reading level, cognitive ability, personality profiles, etc. Furthermore, the data only include measures related to depression, whereas other mental health issues such as anxiety, post-traumatic stress disorder, stress, etc., are not covered. Therefore, we cannot make any claims as to whether the findings can be generalised to any disorders besides depression.

## Other remarks

The use of open-ended text responses requires other demands besides rating scales. For example, as open-ended questions are more demanding, less people tend to answer them [24]. Furthermore, the type of questions matter, where comment-specific questions (e.g., "I feel happy to meet her") yield higher response rates than explanatory-specific questions (e.g., "I meet her because she makes me feel happy"). Response rates also depend on demographic variables, where young males with high online literacy are more likely to answer open-ended questions than old females with low online experience [24]. Finally, the format of a free text response box matters, where more lines evoke longer text responses [25].

## Conclusion

Our findings suggest that respondents prefer free text responses because they, for example, provide better opportunities for elaboration, allowing for precise and natural descriptions of their mental health issues, whereas rating scales are viewed as being faster and quicker. Progress in computational language assessment now provides opportunities to quantify languages with an accuracy that matches respondents' expectation of high precision and validity while using text response as outcome variables in behavioural science (e.g., [2]). The results together suggest that future assessment of mental health patients may benefit from support by computational methods that use open-ended text data.

## Supporting information

**S1 Appendix.**
(DOCX)

## Author Contributions

**Conceptualization:** Sverker Sikström, Oscar Kjell.

**Data curation:** Sverker Sikström, Alfred Pålsson Höök.

**Formal analysis:** Sverker Sikström.

**Funding acquisition:** Sverker Sikström, Oscar Kjell.

**Investigation:** Sverker Sikström, Alfred Pålsson Höök.

**Methodology:** Sverker Sikström, Alfred Pålsson Höök, Oscar Kjell.

**Project administration:** Sverker Sikström.

**Software:** Sverker Sikström.

**Supervision:** Sverker Sikström.

**Writing – original draft:** Sverker Sikström.

**Writing – review & editing:** Sverker Sikström, Oscar Kjell.

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
