## [Decision Letter · Decision Letter 0]

29 Jul 2022

PONE-D-22-11336Precise Language Responses Challenge Easy Rating Scales - Comparing Clinicians’ and Respondents’ ViewsPLOS ONE

Dear Dr. Sikström,

Thank you for submitting your manuscript to PLOS ONE. After careful consideration, we feel that it has merit but does not fully meet PLOS ONE’s publication criteria as it currently stands. Therefore, we invite you to submit a revised version of the manuscript that addresses the points raised during the review process.

We look forward to receiving your revised manuscript.

Kind regards,

Maki Sakamoto, Ph.D

Academic Editor

PLOS ONE

Journal Requirements:

“Marianne och Marcus Wallenbergs Stiftelse (MMW-2021.0058). ”AI-based language models for improving diagnostics, monitoring, and outcome of depression and anxiety”.

Vinnova. Förbättrad diagnostisering av mental hälsa med beskrivande ord och artificiell intelligens. (2018-02007).

Kamprad Foundation. Förbättrad diagnostik för psykisk ohälsa hos äldre: implementering av beslutsstöd baserat på beskrivande ord och artificiell intelligens, ref # 20180281”

“Sverker Sikström and Oscar Kjell are shareholders of WordDiagnostic AB”

Reviewers' comments:

Reviewer's Responses to Questions

**Comments to the Author**

1. Is the manuscript technically sound, and do the data support the conclusions?

Reviewer #1: Partly

Reviewer #2: Yes

2. Has the statistical analysis been performed appropriately and rigorously? 

Reviewer #1: Yes

Reviewer #2: I Don't Know

3. Have the authors made all data underlying the findings in their manuscript fully available?

Reviewer #1: Yes

Reviewer #2: No

4. Is the manuscript presented in an intelligible fashion and written in standard English?

Reviewer #1: Yes

Reviewer #2: Yes

5. Review Comments to the Author

Reviewer #1: I read this manuscript with a lot of interest. Being a researcher and clinician, I am always interested in methods that may improve assessment accuracy and validity. The authors of this study suggest that allowing respondents to express their experience using natural language reflect their experiences with more precision and elaboration. The authors also suggest that this method does not compromise diagnostic accuracy. They also conclude that when compared to clinicians, individuals who participated in the study preferred communicating their experiences in an open text format as opposed to closed format.

This paper opens a discussion for clinicians and researchers and raises important questions that may contribute to literature and advance knowledge. The application of their methodology to clinical samples is also novel and warrants further research and development.

However, I have several fundamental questions about the conclusions that authors draw from the results and a series of various issues that I am going to address.

The finding that respondents found the open-ended format responses to be a more precise depiction of their experience is hardly surprising. In clinical settings close ended rating scales are used as screening tools, provide an estimation for symptom severity, and can also be used to gauge progress. However, a diagnosis is never based on a screening tool alone. In general clients undergo a structured or semi structured interview conducted by a clinician who prompt them to describe their experience in their words. Only after careful elaboration a diagnosis can be provided.

Thus, the first fundamental question pertains to the clinical utility that open-ended format questions can offer given that 1. It is already accomplished though the interview part of the diagnostic process (MINI, or semi structured interview that is conducted when diagnosing the client) 2. Authors mention that this method does not seem to offer incremental validity (given the high correlation with validated existing scales). Did authors mean to imply that open ended questions may provide a richer data and a more useful information in research settings but may not necessarily be useful in clinical settings? Do authors suggest that open-ended format questions may be used in clinical settings as they may save on time and resources when no clinician is available to conduct the clinical interview?

My second fundamental question pertains to “precision and accuracy”, the perception that client’s account is precise and accurate may not always be relevant to the diagnostic criteria. For example, in some individuals with mental health conditions, perceptions of reality can be distorted, can include a psychotic process, can be irrelevant and overinclusive. Open ended questions may not be suitable for some individuals with OCD related disorders etc. Why do authors believe that perception of precision adds to incremental validity? Will this information be better assessed during clinical interview by a clinical psychologist or a diagnostician?

The third fundamental question pertains to the process of gathering information in vulnerable populations. For example, Jon Frederickson in his book on dynamic therapy techniques (p205) reminds clinicians that “Getting information without regulating the fragile patient’s anxiety would be harmful, not healing”. In the event where respondents were “reinstating or evoking” emotions, or “thought differently about their experiences” were they provided with interventions or strategies to manage distress, avoid re-traumatizing in case of PTSD or trauma related disorder, manage suicidality etc?

The fourth fundamental questions related to study population; a degree of cognitive impairment has been documented in individuals with MDD. Slower processing speed, memory issues, and other cognitive domains may be affected during depressive episodes. Is it possible that their accounts might not be factually precise?

Other issues I wanted to address briefly:

1. Title: I am not convinced that the title reflects study results. I don’t see how precise language responses challenge the rating scales. Close ended scales are used for a particular purpose and in particular context. They are not intended to capture the precise experiences that are highly individual for each respondent.

2. It is no doubt that short close ended questionnaires have limitations. Some are constrained by limited questions, some do not capture the full symptomatology (for example, PHQ has been shown to fail to identify depressed individuals with dysthymia). However, despite the limitations short screening measures have been shown to have high validity, can be normed, and using methods such as ROC can be used to establish meaningful cut-offs. Can authors discuss ways that NLP methodology using the open-ended questionnaires can used in a way that captures the severity of symptoms in a clinically meaningful way (cut-offs, severity, norms etc).

3. Short format close ended scales can also be advantageous as they can capturing most domains of symptomatology (somatic, cognitive, behavioural). For example, individuals with MDD may not fully be aware that lack of appetite, difficulty concentrating and decreased sex drive can be part of depression and thus not mention this in their description. Individual with MDD may also have difficulty retrieving information from memory and have slower mental processing speed. It is possible that some of these individuals will not be able to retrieve information that is pertinent. Short close ended scales such as BDI captures these aspects in case these will be missed. Can authors discuss how to address the issue of information that may be missed or underreported, but nonetheless be an important part of diagnosis.

4. It appears that this method will require an exclusion criteria ( i.e, low IQ, comorbid health conditions, psychotic processes, PTSD, cognitive impairment due to MDD or other conditions etc). Will that be a significant limitation?

5. Can authors comment on cultural factors that may affect the validity of their methodology ( more somatic presentations in some populations, non-disclosure etc)

6. In introduction the authors reference a study that was not published yet, to suggest that their method was validated by SADS, and is similar to results obtained through MINI or equivalent semi structured or structured interview. If SADS and MINI accomplish the same goal what advantage does NLP methodology offer above and beyond existing measures? Also, is it possible that some of the diagnoses were missed through PHQ and GAD is based on the fact that these are screeners and cannot be used as stand-alone diagnostic tools?

7. I have few comments regarding the choice of words: perhaps instead of “normal” controls, controls can be described as individual who did not meet the criteria for MDD or GAD. Other examples include the word “attitude” that I believe is meant to suggest preference, or perception, or biases (in methods, and discussion components).

8. While this method seems to be relatively new in clinical settings, can authors provide more references where it was evaluated in clinical samples?

Reviewer #2: This study examines participants’ perceptions of the precision and ease of self-reporting their depressive symptoms using different forms of assessment: traditional rating scales (i.e., PHQ-9), selecting 5 of 30 predetermined words (based off a previous study on assessing depressive symptoms by the same authors), self-selecting 5 words of the participant’s choice, and free text (minimum of 20 words). The findings reported in the study reveal that participants found the free text response form to be largely more precise and easier than traditional rating scales; however, they did not prefer free text over rating scales. Clinicians were generally able to correctly predict what participants would respond but overestimated participants’ perceived difficulty with and dislike for free text responses. The authors based this work on previous studies by their research team on using natural language processing and machine learning algorithms to analyze free text responses describing depressive symptoms, thereby demonstrating the potential relevance and utility of this form of self-report measure over and above traditional rating scales.

Strengths of the current study include incorporating effect sizes throughout the statistical analyses, clarity and accessibility of the explanations on specialized topics (e.g., QCLA, NLP), and clarity (i.e., use of italics) in describing the 12 dimensions of perceptions towards the response options.

Weaknesses of the current study include incomplete descriptions of many aspects of the study (e.g., sample, data analyses, key clinical implications) as well as the lack of clarity surrounding the ethics approval or choice not to share data for this study.

Overall, this study shows great promise in expanding upon and improving current self-report assessments from depression, which could have meaningful implications in research and clinical contexts. However, there are some major concerns regarding how this study was ethically conducted. There are also a few crucial points to address and areas that could be expand upon before this manuscript meets an acceptable level of scientific rigor and demonstrate a thorough understanding of the topic at hand. I believe that, with some major revisions, this manuscript might be acceptable for publication in PLOS One.

1. I have strong concerns with the fact that it doesn’t appear that this study received approval from a research ethics board. Generally, studies conducted with human subjects should undergo an ethics process. In addition, this study asks ‘respondents’ to rate their mental health symptoms, suggesting that there is potential risk. In fact, the 9th item of the PHQ-9 assesses suicidality, which has been shown to elevate risk of suicidal ideation for some people.

2. Greater clarity is needed regarding the participant sample. First of all, the choice to include participants with GAD seems strange, given that the study assessed for depressive symptoms. In fact, the authors state on page 2 that, “similar constructs, such as depression and anxiety, tend to correlate strongly with rating scales, whereas they are better differentiated by QCLA (Kjell et al, 2019).” This suggests that ‘respondents’ with GAD might have different perceptions of completing assessments, especially free text responses, of depressive symptoms. Also unclear is the choice of nurses, psychologists, and medical doctors to represent clinicians and the choice to include participants who did not report having MDD or GAD. More clearly stating the inclusion and exclusion criteria for participants might help the reader better understand the authors’ choices on these points.

3. The manuscript would benefit greatly from the inclusion of a data analysis section to walk the reader through the statistic process of the study’s analyses. This should include a clarification of the choice of when to correct for multiple comparisons.

4. Adding a section in the results to characterize the sample, especially with regards to the severity of mental health symptoms, would greatly enhance the paper. For example, it is unknown to the reader whether the sample is characterized by mild, moderate, or severe levels of depression.

5. Although it is interesting that the authors performed subgroup analyses to compare these 3 groups within the sample, the authors did not explain what the goal of these comparisons was. The relevance of clinicians’ perceptions of their patients’ perceptions of the precision, ease, and preference towards different rating forms for depression is unclear.

6. The language regarding the clinician responses to survey items is confusing throughout the text, as it is often directly compared to ‘respondent’ responses. For example, the note for Figure 2A states, “Figure 2A shows respondents minus clinicians where a positive value indicates that respondents prefer free text responses over rating scales more than clinicians do.” This implies that the figure is comparing the respondent preferences to clinician preferences, despite that clinician preferences are not assessed.

7. Given that the purpose of this study appears to be to demonstrate the potential utility and relevance of using free text responses to assess depressive symptoms, the authors might want to consider reporting on the psychometric properties of their measures to whatever extent is possible. A comparison of the psychometric properties between a validated measure of depression (i.e., the PHQ-9) and other forms of assessing depression might also be relevant. The argument can be made that, if the other forms of assessing depression are not reliable and/or valid, then the perceptions of ‘respondents’ and clinicians is irrelevant. Statements in the manuscript about the accuracy of text data for assessing mental health should be removed, as the study data do not support this conclusion.

8. This study’s relevance is strongly related to the potential of applying question-based computational language assessment to research and clinical contexts. However, these are hardly discussed in the introduction and discussion sections of the paper. This would also give the authors an opportunity to elaborate on the likelihood of such resource-intensive assessments to be taken up by, for example, clinics not associated with academic institutions.

9. The conclusion statement does not summarize the findings of the current study in a nuanced manner, focusing solely on ‘elaboration’ rather than including the other 11 dimensions of participant perceptions.

Minor:

1. Although minor, there are some grammatical and phrasing issues that impact clarity throughout the text, especially in the introduction and discussion sections (e.g., similar rather than similarly on page 2, “primary and secondary symptoms” on page 3, use of “phases” on page 6). A thorough review of the manuscript should address these issues.

2. I would suggest either clarifying the inclusion of the study results from Kjell, Daukantaitë & Sikström, 2021 (page 3) or removing this reference as it does not pertain to mental health assessments.

3. The clarity of the study would be improved by placing the “materials” section after the “procedure” section; this way, the reader can contextualize the different response formats.

4. Please clarify the Likert scale presented on page 6, as the response options do not match the questions included in the Appendix.

5. Reference to Table 1b should be moved to the ‘clinician condition’ section in the results section.

6. PLOS authors have the option to publish the peer review history of their article (what does this mean?). If published, this will include your full peer review and any attached files.

Reviewer #1: No

Reviewer #2: **Yes: **Catherine Paré

---

## [Author Response · Author response to Decision Letter 0]

8 Oct 2022

Rebuttal letter of for ”Precise Language Responses Challenge Easy Rating Scales - Comparing Clinicians’ and Respondents’ Views” (PONE-D-22-11336) PLOS ONE

Dear editor and reviewers, thank for you’re the insights full comments to our manuscript (PONE-D-22-11336). Please find below our answers to your comments.

Journal Requirements:

Our response: We have added the following sentence at the end of the ethical section

“Participant were required to give written informed consent of their participations in the study prior to starting the study. All participants were adults (i.e., above 18 years). The approval of the consent was documented in the response data file. All data was collected anonymously.”

“Marianne och Marcus Wallenbergs Stiftelse (MMW-2021.0058). ”AI-based language models for improving diagnostics, monitoring, and outcome of depression and anxiety”.

Vinnova. Förbättrad diagnostisering av mental hälsa med beskrivande ord och artificiell intelligens. (2018-02007).

Kamprad Foundation. Förbättrad diagnostik för psykisk ohälsa hos äldre: implementering av beslutsstöd baserat på beskrivande ord och artificiell intelligens, ref # 20180281”

Our response: We have added the following sentence in the Funding section. 

“Sverker Sikström and Oscar Kjell are shareholders and founders of WordDiagnostics AB”

Our response: We have added the following statement

“This does not alter our adherence to PLOS ONE policies on sharing data and materials.”

Reviewers' comments:

5. Review Comments to the Author

Reviewer #1: I read this manuscript with a lot of interest. Being a researcher and clinician, I am always interested in methods that may improve assessment accuracy and validity. The authors of this study suggest that allowing respondents to express their experience using natural language reflect their experiences with more precision and elaboration. The authors also suggest that this method does not compromise diagnostic accuracy. They also conclude that when compared to clinicians, individuals who participated in the study preferred communicating their experiences in an open text format as opposed to closed format.

This paper opens a discussion for clinicians and researchers and raises important questions that may contribute to literature and advance knowledge. The application of their methodology to clinical samples is also novel and warrants further research and development.

Our response: Thank you. However, we would like to note that this study is about respondents’ and clincians’ preference of CLA, whereas other articles/manuscripts from our research group is about evidence for the validity and clinical use of CLAs and thus not the focus of this manuscript.

However, I have several fundamental questions about the conclusions that authors draw from the results and a series of various issues that I am going to address.

The finding that respondents found the open-ended format responses to be a more precise depiction of their experience is hardly surprising. In clinical settings close ended rating scales are used as screening tools, provide an estimation for symptom severity, and can also be used to gauge progress. However, a diagnosis is never based on a screening tool alone. In general clients undergo a structured or semi structured interview conducted by a clinician who prompt them to describe their experience in their words. Only after careful elaboration a diagnosis can be provided.

Thus, the first fundamental question pertains to the clinical utility that open-ended format questions can offer given that 1. It is already accomplished though the interview part of the diagnostic process (MINI, or semi structured interview that is conducted when diagnosing the client) 2. Authors mention that this method does not seem to offer incremental validity (given the high correlation with validated existing scales). Did authors mean to imply that open ended questions may provide a richer data and a more useful information in research settings but may not necessarily be useful in clinical settings? Do authors suggest that open-ended format questions may be used in clinical settings as they may save on time and resources when no clinician is available to conduct the clinical interview?

Our response: In later part of the introduction, we write:

“We argue that computational assessment of language responses to health-related questions is not typically done in clinical settings, and that this method can improve the validity of assessment. Interviews are part of the current clinical assessment methods (e.g., MINI Sheehan et al., 1998), however, current practice does not use computational method for this assessment. Previous studies provide evidence that computational assessment of language data can improve the validity of the assessment above rating scales, for example better categorization of facial expressions (Kjell et al., 2019), and better correlations with cooperative behavior (Kjell, Daukantaite, Sikström, 2021). These data suggest that computational assessment of language data may potentially improve the validity of assessment of MDD and GAD in research and clinical settings. However, further research is needed to support this claim. Less data is available on direct comparisons between how well computational assessment of text data on health questionnaires compares with experienced clinical evaluations of the same text data. Beyond the question of validity, computational assessment of text data can also provide other advantages as it can be used when there is no clinical available to conduct a clinical interview, provide a second opinion to clinical interviews, and provides a richer data than rating scales alone.”

My second fundamental question pertains to “precision and accuracy”, the perception that client’s account is precise and accurate may not always be relevant to the diagnostic criteria. For example, in some individuals with mental health conditions, perceptions of reality can be distorted, can include a psychotic process, can be irrelevant and overinclusive. Open ended questions may not be suitable for some individuals with OCD related disorders etc. Why do authors believe that perception of precision adds to incremental validity? Will this information be better assessed during clinical interview by a clinical psychologist or a diagnostician?

Our response: In the limitation section we argue that:

“We are not claiming that all assessment of mental health is well supported by computational language assessment. This article focuses on depression, and prior work from our lab has also successfully included studies on anxiety, harmony in life, and satisfaction with life (e.g., Kjell et al., 2019). Common to these assessments is that they relate to the emotional and/or cognitive states/experiences of the participants, where such states and experiences are commonly communicated in language. However, in our view, language based on assessment methods may be less suitable for mental health issues related to cognitive performance, for example ADHD or dementia. Furthermore, how well computational language assessment methods is suitable for psychotic disorders or mental health issues related to anxiety, or depression, for example obsessive compulsive disorder (OCD) is a topic for future research. “

The third fundamental question pertains to the process of gathering information in vulnerable populations. For example, Jon Frederickson in his book on dynamic therapy techniques (p205) reminds clinicians that “Getting information without regulating the fragile patient’s anxiety would be harmful, not healing”. In the event where respondents were “reinstating or evoking” emotions, or “thought differently about their experiences” were they provided with interventions or strategies to manage distress, avoid re-traumatizing in case of PTSD or trauma related disorder, manage suicidality etc.?

Our response: In the later part of the discussion, we have added a section of how writing about emotional events may influence the severity of mental health:

“Writing about mental health issues is not only a method of assessment, but it also influences the severity of mental health. A large body of empirical evidence shows that repeated writing about mental health improve the severity of mental health issues such as anxiety and depression (Gerger et al., 2021), whereas the improvement following mental health rating scales is modest (Bastiaansen et al., 2020). Pennebaker and Beall (1986) originally suggested the expressive writing paradigm, where four episodes of 15 minutes writing of a traumatic episode diminished the severity of posttraumatic stress disorder (PTSD). However, re-exposure of traumatic events should also be done with care. Frederickson (2013) argued that “getting information without regulating the fragile patient’s anxiety would be harmful, not healing” and reminded clinicians that re-exposures interventions should be combined with strategies of manage distress related to traumatic event (p. 205). However, a recent meta review of the expressive writing paradigm suggests that the improvement in mental health related to, for example depression, is a stable and a reproducible phenomenon (Gerger et al., 2021). “

Bastiaansen, JA. Ornée DA., Meurs M., Oldehinke AJ. (2020). An evaluation of the efficacy of two add-on ecological momentary intervention modules for depression in a pragmatic randomized controlled trial (ZELF-i). 

Frederickson F. (2013). Co-Creating Change: Effective Dynamic Therapy Techniques. ISBN 0988378841. Seven Leaves Press

Pennebaker, J. W., & Beall, S. K. (1986). Confronting a traumatic event: Toward an understanding of inhibition and disease. Journal of Abnormal Psychology, 95, 274–281.

Gerger H, Werner CP, Gaab J, Cuijpers P (2021). Comparative efficacy and acceptability of expressive writing treatments compared with psychotherapy, other writing treatments, and waiting list control for adult trauma survivors: a systematic review and network meta-analysis. Psychological Medicine 1–13. https://doi.org/10.1017/ S0033291721000143 

The fourth fundamental questions related to study population; a degree of cognitive impairment has been documented in individuals with MDD. Slower processing speed, memory issues, and other cognitive domains may be affected during depressive episodes. Is it possible that their accounts might not be factually precise?

Our response: In the end of the discussion, we have added

“MDD is associated with cognitive impairment such as slower processing speed and memory impairments (Lam et al., 2014). A related question here is what the extent such cognitive impairments may lower the validity of MDD patients’ text responses analyzed by computational methods. To our knowledge, this question has not been carefully analyzed. However, a related study by Sikström et al. (in progress) indicates that the elderly people, that also is associated with cognitive decline in memory and processing speed, show higher validity in computational assessment of language responses than young people. Further research is needed to better understand the relation between cognitive impairment in MDD and computational assessment of language responses.“ 

Lam RW, Kennedy SH, Mclntyre RS, Khullar A. Cognitive dysfunction in major depressive disorder: effects on psychosocial functioning and implications for treatment. Can J Psychiatry. 2014 Dec;59(12):649-54. doi: 10.1177/070674371405901206. PMID: 25702365; PMCID: PMC4304584.

Sikström, S., Kelmendi B. (in progress). Assessment of Depression and Anxiety in Young and Old with Question Based Computational Language Model.

Other issues I wanted to address briefly:

1. Title: I am not convinced that the title reflects study results. I don’t see how precise language responses challenge the rating scales. Close ended scales are used for a particular purpose and in particular context. They are not intended to capture the precise experiences that are highly individual for each respondent.

Our response: We have replaced the word “challenging” in title with “versus” so that the updated title is “Precise Language Responses versus Easy Rating Scales - Comparing Clinicians’ and Respondents’ Views”. 

2. It is no doubt that short close ended questionnaires have limitations. Some are constrained by limited questions, some do not capture the full symptomatology (for example, PHQ has been shown to fail to identify depressed individuals with dysthymia). However, despite the limitations short screening measures have been shown to have high validity, can be normed, and using methods such as ROC can be used to establish meaningful cut-offs. Can authors discuss ways that NLP methodology using the open-ended questionnaires can used in a way that captures the severity of symptoms in a clinically meaningful way (cut-offs, severity, norms etc).

Our response: In the end of the discussion, we have added:

“Rating scales have been a standard method to measure mental states in psychological research for the last century. Several standard methods to take advantage of, understand, and apply rating scales is currently in use. Examples of such methods are Receiver Operating Curves (ROC, McNicol, 2004), Signal Detection Theory (SDT), and Item Response Theory (IRT, Thissen & Steinberg, 1988). Computational language assessment allows us to map freely generated texts to a single quantifiable dimension associated with, for example, a rating scale. This can be achieved by training a machine learning method take a set of texts and generate a scale that correlates well with an associated rating scale (e.g., see Kjell et al., 2019; 2021). This semantic scale can then be used as a severity measure, where cut-offs can be used to categorize the text response into mild, moderate, or severe MDD or GAD. In addition, the generated unidimensional semantic scale can then be applied to the methods developed for rating scales (e.g., ROC, SDT, IRT). Further research is needed to carefully analyses possibilities and limitations while applying rating scales methods to semantic scales.“

Thissen, D., Steinberg, L. (1988). Data analysis of using the item response theory. Psychological Bulletin, 104 (3), 385-395. 

McNicol, D. (2004). A primer of signal detection. Psychology Press. https://doi.org/10.4324/9781410611949

3. Short format close ended scales can also be advantageous as they can capturing most domains of symptomatology (somatic, cognitive, behavioural). For example, individuals with MDD may not fully be aware that lack of appetite, difficulty concentrating and decreased sex drive can be part of depression and thus not mention this in their description. Individual with MDD may also have difficulty retrieving information from memory and have slower mental processing speed. It is possible that some of these individuals will not be able to retrieve information that is pertinent. Short close ended scales such as BDI captures these aspects in case these will be missed. Can authors discuss how to address the issue of information that may be missed or underreported, but nonetheless be an important part of diagnosis.

Our response: This topic has been discussed in the end of the discussion section:

“The proposed method of question-based computational language assessment may be based on general mental health questions (i.e., describe your mental health) or more specific questions related to symptoms (describe your appetite, concentration, etc.). General questions have the advantage of not priming, or activating, participant to focus on aspects that may not be relevant for a particular patient, whereas specific questions may help participants to focus on aspect that they otherwise would have missed. Thus, general, and specific questions may have complementary purposes, where general questions should proceed specific questions. Previous studies (Kjell et al., 2021) have shown that computational assessment of all open-ended questions related to specific MDD symptoms correlates with standard rating scales of depression (i.e., PHQ-9); however, cognitive, and emotional aspects yielded higher predictability than secondary criteria of behavioral aspects. Furthermore, combining several open-ended questions improves accuracy compared to the single questions (Kjell, Sikström et al., 2021). More importantly, as computational language assessment is based on machine learning, it may pick up relevant cues that are implicitly related to the to-be-measured construct. For example, the excessive usage of the first pronouns in freely generated texts (i.e. “I”) is associated with depression, although the text may not necessary be related to mental health (Rude et al., 2004).” 

Rude, S. S,. Gortner E-M, Pennebaker JW. (2004). Language Use of Depressed and Depression-Vulnerable College Students. Cognition and Emotion 18(8):1121-1133 Follow journal. DOI: 10.1080/02699930441000030

4. It appears that this method will require an exclusion criteria (i.e, low IQ, comorbid health conditions, psychotic processes, PTSD, cognitive impairment due to MDD or other conditions etc). Will that be a significant limitation?

Our response: Please see our response to the fundamental questions number 2 and 4.

5. Can authors comment on cultural factors that may affect the validity of their methodology (more somatic presentations in some populations, non-disclosure etc)

Our response: This topic has been discussed in the limitation section:

“Cultural factors may influence assessment of depression. This may be particularly relevant for computational language assessments as different populations may use different languages. Although this topic has not been systematically investigated, studies of computational language assessment of young and old may provide hints, where training of data on old population generalized well to a young population and vice versa, however the former training was significantly more successful than later (Sikström & Kelmendi., in progress).“

6. In introduction the authors reference a study that was not published yet, to suggest that their method was validated by SADS, and is similar to results obtained through MINI or equivalent semi structured or structured interview. If SADS and MINI accomplish the same goal what advantage does NLP methodology offer above and beyond existing measures? 

Our response: In this study we used an online (rating scale) version of MINI that we label SDAS. To make this text clearer, and without going into technical details, we have removed to reference to the MINI here. 

Also, is it possible that some of the diagnoses were missed through PHQ and GAD is based on the fact that these are screeners and cannot be used as stand-alone diagnostic tools?

Our response: This is exactly the point that we are making here, and this has now been clarified by adding this sentence

“This suggest that the validity of the PHQ-9 or the GAD-7 may be improved by adding open-ended questions assessed by computational language assessment. “

7. I have few comments regarding the choice of words: perhaps instead of “normal” controls, controls can be described as individual who did not meet the criteria for MDD or GAD. Other examples include the word “attitude” that I believe is meant to suggest preference, or perception, or biases (in methods, and discussion components).

Our response: Following the reviewer’s suggestion we now describe the control group as the “respondents who did not report these diagnoses, hereafter referred to as the control group.” Furthermore, we have replaced the concept of “attitudes” to “preference” throughout the manuscript. 

8. While this method seems to be relatively new in clinical settings, can authors provide more references where it was evaluated in clinical samples?

Our response: We would like to emphasize that our paper is not about validation of QCLA, it is about the patients’ and clinicians’ preference of response formats. Nevertheless, we have two ongoing projects where the computational language assessment method is being 1) validated against predicting the M.I.N.I. as being carried out by an experienced clinical psychologist; and 2) validated against best estimate assessment using the Longitudinal Expert All Data method. We have added the following sentences:

”Further, Sikström et al. (in progress) showed that a combination of word responses and rating scales diminishes the number of misses in diagnosis (keeping false alarms constant) of depression and anxiety by approximately half compared to state-of-the-art rating scales (the Patient Health Questionnaire, PHQ-9, [Kroenke, Spitzer, Williams 2001] and the Generalized Anxiety Disorder Scale, GAD-7 [Spitzer, Kroenke, Williams, & Löwe, 2006]) when validated by criteria related to DSM-5 (American Psychiatric Association, 2013). This suggest that the validity of the PHQ-9 or the GAD-7 may be improved by adding open-ended questions assessed by computational language assessment.”

Reviewer #2: This study examines participants’ perceptions of the precision and ease of self-reporting their depressive symptoms using different forms of assessment: traditional rating scales (i.e., PHQ-9), selecting 5 of 30 predetermined words (based off a previous study on assessing depressive symptoms by the same authors), self-selecting 5 words of the participant’s choice, and free text (minimum of 20 words). The findings reported in the study reveal that participants found the free text response form to be largely more precise and easier than traditional rating scales; however, they did not prefer free text over rating scales. Clinicians were generally able to correctly predict what participants would respond but overestimated participants’ perceived difficulty with and dislike for free text responses. The authors based this work on previous studies by their research team on using natural language processing and machine learning algorithms to analyze free text responses describing depressive symptoms, thereby demonstrating the potential relevance and utility of this form of self-report measure over and above traditional rating scales.

Strengths of the current study include incorporating effect sizes throughout the statistical analyses, clarity and accessibility of the explanations on specialized topics (e.g., QCLA, NLP), and clarity (i.e., use of italics) in describing the 12 dimensions of perceptions towards the response options.

Weaknesses of the current study include incomplete descriptions of many aspects of the study (e.g., sample, data analyses, key clinical implications) as well as the lack of clarity surrounding the ethics approval or choice not to share data for this study.

Overall, this study shows great promise in expanding upon and improving current self-report assessments from depression, which could have meaningful implications in research and clinical contexts. However, there are some major concerns regarding how this study was ethically conducted. There are also a few crucial points to address and areas that could be expand upon before this manuscript meets an acceptable level of scientific rigor and demonstrate a thorough understanding of the topic at hand. I believe that, with some major revisions, this manuscript might be acceptable for publication in PLOS One.

1. I have strong concerns with the fact that it doesn’t appear that this study received approval from a research ethics board. Generally, studies conducted with human subjects should undergo an ethics process. In addition, this study asks ‘respondents’ to rate their mental health symptoms, suggesting that there is potential risk. In fact, the 9th item of the PHQ-9 assesses suicidality, which has been shown to elevate risk of suicidal ideation for some people.

Our response: The study, its methods, and the fact that we have not applyed for ethical approval for this exact study follow Swedish law regarding research and ethics: This is because this type of research does not require ethical approval according to Swedish law. In fact, we are sure about this because we have applied for ethical approval from the Swedish national research ethics board for a very similar, but riskier, set of studies including the same population, and similar research questions and tools (e.g., including the PHQ-9, the GAD-7, the PSS-10, and these open-ended questions in addition to questions about depression, worry etc.). The national ethics board returned that ethics application stating that: The study was deemed exempt from requiring ethical approval according to Swedish Law (see §§ 3-4 of the Act [2003:460] on ethical review of research involving humans in Sweden). This is because the research involved anonymous data, and a procedure that was considered to not include an “obvious” (uppenbar) risk for physical or psychological harm (i.e., the research did not fall within this paragraph: “Forskningen utförs enligt en metod som syftar till att påverka forskningspersonen fysiskt eller psykiskt, eller så innebär forskningen en uppenbar risk att skada forskningspersonen.”). It is possible to receive this ethics application and its decision by emailing registrator@etikprovning.se, providing the ID 2020-00730 (for more information about the Swedish research ethics application system please see www.epn.se).

2. Greater clarity is needed regarding the participant sample. First of all, the choice to include participants with GAD seems strange, given that the study assessed for depressive symptoms. In fact, the authors state on page 2 that, “similar constructs, such as depression and anxiety, tend to correlate strongly with rating scales, whereas they are better differentiated by QCLA (Kjell et al, 2019).” This suggests that ‘respondents’ with GAD might have different perceptions of completing assessments, especially free text responses, of depressive symptoms. Also unclear is the choice of nurses, psychologists, and medical doctors to represent clinicians and the choice to include participants who did not report having MDD or GAD. More clearly stating the inclusion and exclusion criteria for participants might help the reader better understand the authors’ choices on these points.

Our response: We have now clarified that:

“The clinician condition included participants who stated they were employed as a nurse, psychologist, or medical doctor. There were 92 participants on Proflic in this group, where 62 of these participants started the study and 40 completed it. This group was chosen because we wanted to target professionals that work with patients in their daily work life, and compare them with a respondent group that did not work professionally with mental health. 

In the respondents’ condition (N = 304), a pre-screening was conducted to find a population where approximately half of the respondents self-reported that they had an ongoing major depression diagnosis (MDD) or generalized anxiety disorder (GAD), and the other half were respondents who did not report these diagnoses, hereafter referred to as the control group. We chose to include participant with MDD or GAD as this is the group of participants that we been systematically investigating in several previous studies (e.g., Kjell et al., 2019; Kjell et al, 2021; Kjell, Johansson, Sikström, 2021). A drawback of this choice was a smaller group of the participants had only GAD diagnose (N = 21), whereas most of them had MDD (N = 145 and sometimes also GAD). 

3. The manuscript would benefit greatly from the inclusion of a data analysis section to walk the reader through the statistic process of the study’s analyses. This should include a clarification of the choice of when to correct for multiple comparisons.

Our response: The following paragraph has been added

“Data analysis. The data was divided into the clinician condition and the respondent condition, and then further divided into the four response categories (rating scales, selected words, freely generated words, and free text) and 12 preferences rating scales related to the response categories. Descriptive analysis was then conducted by reporting the mean responses of each of the rating scales for each response category. Inferential statistic was made using t-tests, with a focus on comparing preferences to the rating scales with the preferences to the free text responses. We used Bonferroni correction for multiple testing of the rating scales (N = 12), and the alpha level was set to .05.”

4. Adding a section in the results to characterize the sample, especially with regards to the severity of mental health symptoms, would greatly enhance the paper. For example, it is unknown to the reader whether the sample is characterized by mild, moderate, or severe levels of depression.

Our response: To characterize the severity of the respondents, the following paragraph has been added to the manuscript:

“PHQ-9 severity scores. The PHQ-9 scores (ranging from 0 to 27) were calculated for the respondents and classified into none-minimal (N = 52, range 0-4), mild (N = 57, range 5-9), moderate (N = 67, range 10-14), moderately severe (N = 69, range 15-19), and severe (N = 59, range 20-27).”

5. Although it is interesting that the authors performed subgroup analyses to compare these 3 groups within the sample, the authors did not explain what the goal of these comparisons was. The relevance of clinicians’ perceptions of their patients’ perceptions of the precision, ease, and preference towards different rating forms for depression is unclear.

Our response: In the research question paragraph, we have added:

”Furthermore, we wanted to compare the preferences of a population respondents taking mental health tests (i.e., patients or non-patients that may seek assessment) with clinicians’ (that select and administrates the assessments) meta-believes of the respondents’ preferences. We were particularly interested in whether the clinicians beliefes of the respondents’ preferences matched the real preferences rated by the respondents. This is an important question as the clinicians should acknowledge respondents’ preferences while selecting the assessment instruments to establish an efficient communication.” 

6. The language regarding the clinician responses to survey items is confusing throughout the text, as it is often directly compared to ‘respondent’ responses. For example, the note for Figure 2A states, “Figure 2A shows respondents minus clinicians where a positive value indicates that respondents prefer free text responses over rating scales more than clinicians do.” This implies that the figure is comparing the respondent preferences to clinician preferences, despite that clinician preferences are not assessed.

Our response: Thank you for pointing this out. We have now clarified that the clinicians’ task was to estimate the respondents’ preferences. For example, Figure 2A captions have been changed to:

“Respondents’ preferences of free text responses over rating scales using clinician estimate of the respondents’ preference as the baseline”

And

“Figure 2A shows respondents’ ratings minus clinicians’ ratings, where a positive value indicates that respondents’ preference of free text responses over rating scales more than clinicians’ estimate of respondents’ preferences.”

7. Given that the purpose of this study appears to be to demonstrate the potential utility and relevance of using free text responses to assess depressive symptoms, the authors might want to consider reporting on the psychometric properties of their measures to whatever extent is possible. A comparison of the psychometric properties between a validated measure of depression (i.e., the PHQ-9) and other forms of assessing depression might also be relevant. The argument can be made that, if the other forms of assessing depression are not reliable and/or valid, then the perceptions of ‘respondents’ and clinicians is irrelevant. Statements in the manuscript about the accuracy of text data for assessing mental health should be removed, as the study data do not support this conclusion.

Our response: To our knowledge, the manuscript does not present data supporting that computers language assessment of mental health has high validity, however, the manuscript references other studies that makes this claim. In the conclusion we have added a missing reference showing that this claim is coming from other studies. Additional psychometric properties of computerized language assessment can be found in other publications and is out of focus in the current manuscript. 

“Progress in computational language assessment now provides opportunities to quantify languages with an accuracy that matches respondents' expectation of high precision and validity while using text response as outcome variables in behavioural science (e.g., Kjell et al., 2019)”. 

8. This study’s relevance is strongly related to the potential of applying question-based computational language assessment to research and clinical contexts. However, these are hardly discussed in the introduction and discussion sections of the paper. This would also give the authors an opportunity to elaborate on the likelihood of such resource-intensive assessments to be taken up by, for example, clinics not associated with academic institutions.

Our response: This has been added in the introduction, for details see our responses from Reviewers 1 comments. 

9. The conclusion statement does not summarize the findings of the current study in a nuanced manner, focusing solely on ‘elaboration’ rather than including the other 11 dimensions of participant perceptions.

Our response: The conclusion in the abstract have been added with the following sentence: 

”Participants have preferences for open response format as it allows them to elaborate, be precise, etc. of their mental health issues whereas rating scales are viewed as faster and easier.”

And this sentence has been added in the conclusion at the end of the manuscript:

“Our finding suggests that respondents prefer free text responses because they, for example, provide better opportunities for elaboration allowing for precise and natural description of their mental health issues, whereas rating scales are viewed as being faster and quicker. “

Minor:

1. Although minor, there are some grammatical and phrasing issues that impact clarity throughout the text, especially in the introduction and discussion sections (e.g., similar rather than similarly on page 2, “primary and secondary symptoms” on page 3, use of “phases” on page 6). A thorough review of the manuscript should address these issues.

Our response: The grammatical errors have been corrected. 

2. I would suggest either clarifying the inclusion of the study results from Kjell, Daukantaitë & Sikström, 2021 (page 3) or removing this reference as it does not pertain to mental health assessments.

Our response: This reference has been removed.

3. The clarity of the study would be improved by placing the “materials” section after the “procedure” section; this way, the reader can contextualize the different response formats.

Our response: This has been done.

4. Please clarify the Likert scale presented on page 6, as the response options do not match the questions included in the Appendix.

Our response: The short presentation of the Likert scale on page 6 now better matches the full phrasing in the appendix in terms of content and order.

5. Reference to Table 1b should be moved to the ‘clinician condition’ section in the results section.

Our response: Done.

---

## [Decision Letter · Decision Letter 1]

15 Nov 2022

PONE-D-22-11336R1Precise Language Responses versus Easy Rating Scales - Comparing Clinicians’ and Respondents’ ViewsPLOS ONE

Dear Dr. Sikström,

Thank you for submitting your manuscript to PLOS ONE. After careful consideration, we feel that it has merit but does not fully meet PLOS ONE’s publication criteria as it currently stands. Therefore, we invite you to submit a revised version of the manuscript that addresses the points raised during the review process.

We look forward to receiving your revised manuscript.

Kind regards,

Maki Sakamoto, Ph.D

Academic Editor

PLOS ONE

Journal Requirements:

Reviewers' comments:

Reviewer's Responses to Questions

**Comments to the Author**

1. If the authors have adequately addressed your comments raised in a previous round of review and you feel that this manuscript is now acceptable for publication, you may indicate that here to bypass the “Comments to the Author” section, enter your conflict of interest statement in the “Confidential to Editor” section, and submit your "Accept" recommendation.

Reviewer #2: All comments have been addressed

2. Is the manuscript technically sound, and do the data support the conclusions?

Reviewer #2: Yes

3. Has the statistical analysis been performed appropriately and rigorously? 

Reviewer #2: Yes

4. Have the authors made all data underlying the findings in their manuscript fully available?

Reviewer #2: No

5. Is the manuscript presented in an intelligible fashion and written in standard English?

Reviewer #2: Yes

6. Review Comments to the Author

Reviewer #2: I want to thank the authors for the time and effort put into addressing the previous comments from myself and other reviewers. The additions and changes made have significantly improved the text, such as the inclusion of a section on Ethics as well as elaboration on discussion points. However, there are still a few issues that I would like to bring up that, I think, remain important and relevant to address.

Major comments:

I feel that there is still some key information lacking regarding the study sample that would help to situate the reader in their understanding of the study:

- Please provide more information (i.e., evidence from the literature) regarding the authors' interest in looking at the perceptions of clinicians regarding what their patients want. Although there is a huge body of literature in mental health about patient expectations, clinicians do not make clinical decisions solely on their patient's preferences (especially in a context such as assessment).

- Please elaborate further on the authors' decision to include individuals with GAD in the sample. I do not feel that being part of the population in previous studies is an acceptable reason for including these 21 participants.

- Please elaborate on the recruitment of respondents (e.g., how many people met inclusion criteria? How were potential participants chosen? How many people dropped out? When did recruitment stop?)

The inclusion of a paragraph on cultural factors is a really great addition to the paper, and would benefit from being expanded upon. Namely, the authors could mention the issues related to sexism and racism historically associated with machine learning (such as those discussed in The Alignment Problem by Brian Christian) and the consequences these might have for non-WASP individuals assessed using QCLA.

My only other major comment is that I feel the manuscript currently relies heavily on previous research conducted by this research group. I understand that there probably aren't many researchers conducting this kind of research (i.e., on QCLA); however, the manuscript often makes claims or refers do more general domains that would benefit from references outside the research group (such as asking general versus specific open-ended question).

Minor comments:

Please rewrite in the title and abstract to clarify that the study assessed clinicians' perceptions of what respondents preferred, rather than clinician preferences. This is well done in the remainder of the manuscript.

Please clarify what are primary and secondary symptoms in the DSM-5; I am a clinician who regularly uses the DSM-5 and am not familiar with these terms.

I am not sure that I understand why the following statement is included in description of clinician participants: "compare them with a respondent group that did not work professionally with mental health." My understanding of the sample is that there is no inclusion of professionals that have not worked in mental health.

The following statements in the discussion is outside the scope of the current study and should be removed:

- "Together these findings open up for possibilities to use open-ended responses without sacrificing accuracy in measuring different constructs."

- the paragraph before the limitations section

Please clarify how the advances in NLP and QCLA would impact patients rating free text as higher than rating scales (on page 12 of the discussion).

7. PLOS authors have the option to publish the peer review history of their article (what does this mean?). If published, this will include your full peer review and any attached files.

Reviewer #2: No

---

## [Author Response · Author response to Decision Letter 1]

12 Dec 2022

Dear Maki Sakamoto

Please find below our rebuttal letter for the manuscript PONE-D-22-11336R1

“Precise Language Responses versus Easy Rating Scales - Comparing Clinicians’ and Respondents’ Views”.

We have carefully addressed all constructive comments from the Reviewer and believe that the manuscript now is acceptable for publication in PLOS ONE. 

Se our detailed comments below:

Reviewer #2: I want to thank the authors for the time and effort put into addressing the previous comments from myself and other reviewers. The additions and changes made have significantly improved the text, such as the inclusion of a section on Ethics as well as elaboration on discussion points. However, there are still a few issues that I would like to bring up that, I think, remain important and relevant to address.

Major comments:

I feel that there is still some key information lacking regarding the study sample that would help to situate the reader in their understanding of the study:

- Please provide more information (i.e., evidence from the literature) regarding the authors' interest in looking at the perceptions of clinicians regarding what their patients want. Although there is a huge body of literature in mental health about patient expectations, clinicians do not make clinical decisions solely on their patient's preferences (especially in a context such as assessment).

Our response: We have motivated the reason for why looking at response view by adding this in the abstract:

“…however, the respondents’ views as to what best communicates mental states are frequently ignored, which is important for making them comply with assessment.”

Further down in the introduction we have added: 

“To study the respondents’ views of the assessment is important, as a working relation between patients and clinician over time is essential for motivating patients to reveal sensitive information.” 

- Please elaborate further on the authors' decision to include individuals with GAD in the sample. I do not feel that being part of the population in previous studies is an acceptable reason for including these 21 participants.

Our response: We have removed the claim that the choice of looking at GAD is related to samples in previous studies, and replaced it with the argument that GAD and MDD has a large comorbidity:

”We chose to include participants with MDD or GAD as these are two major mental health disorders with a high comorbidity (i.e., rating scales of measuring them correlates highly).”

- Please elaborate on the recruitment of respondents (e.g., how many people met inclusion criteria? How were potential participants chosen? How many people dropped out? When did recruitment stop?)

Our response: We have now specified that 92 participants on Prolific met the inclusion criteria in the clinician group; and that of those 62 started the survey, 40 completed it, and 22 dropped out:

”There were 92 participants on Prolific that met the criteria for taking part in the clinician condition, where 62 of these participants started the study and 40 completed it (22 dropped out prior to completing the study.)”

In the respondents condition we have clarified that: 

”We chose to include participants with MDD or GAD as these are two major mental health disorders with high comorbidity (i.e., rating scales of measuring them correlates highly). Participants that matched the pre-screened condition were invited to participate until the pre-defined number of participants was reached. The data collection lasted for approximately 17 hours on Prolific. Data of participants that did not complete the study was not saved.”

The inclusion of a paragraph on cultural factors is a really great addition to the paper, and would benefit from being expanded upon. Namely, the authors could mention the issues related to sexism and racism historically associated with machine learning (such as those discussed in The Alignment Problem by Brian Christian) and the consequences these might have for non-WASP individuals assessed using QCLA.

My only other major comment is that I feel the manuscript currently relies heavily on previous research conducted by this research group. I understand that there probably aren't many researchers conducting this kind of research (i.e., on QCLA); however, the manuscript often makes claims or refers do more general domains that would benefit from references outside the research group (such as asking general versus specific open-ended question).

We have added a paragraph that refers to more general domains and that discusses the point raised by the Reviewer: 

“Another concern is that machine learning that is used by QCLA may be influenced by biases. For example, women are more likely to experience anxiety, leading to the concern that the machine learning may incorrectly use female language as evidence for anxiety. This effect has been labelled the Alignment Problem (Christian, 2022). Depending on the training data, this problem may influence various ethnical groups, and may be a disadvantage that may be particularly problematic for non-White Anglo-Saxon Protestants (WASP). “

Minor comments:

Please rewrite in the title and abstract to clarify that the study assessed clinicians' perceptions of what respondents preferred, rather than clinician preferences. This is well done in the remainder of the manuscript.

Our response:

The title has been changed to:

” Precise Language Responses versus Easy Rating Scales - Comparing Respondents’ Views with Clinicians’ belief of the Respondent’s Views”

Two sentences have been changed in the abstract:

"This was compared with the clinicians’ (N = 40) belief of the respondent’s view.”

” Respondents preferred the free text responses to a greater degree than rating scales compared to clinicians’ belief of the respondents’ views.”

Please clarify what are primary and secondary symptoms in the DSM-5; I am a clinician who regularly uses the DSM-5 and am not familiar with these terms.

Our response: We have removed the concept of ”primary and secondary” symptoms in DSM-5.

I am not sure that I understand why the following statement is included in description of clinician participants: "compare them with a respondent group that did not work professionally with mental health." My understanding of the sample is that there is no inclusion of professionals that have not worked in mental health.

Our response: This part of the sentence has been removed. 

The following statements in the discussion is outside the scope of the current study and should be removed:

- "Together these findings open up for possibilities to use open-ended responses without sacrificing accuracy in measuring different constructs."

- the paragraph before the limitations section

Our response: This sentence and this paragraph have been removed. 

Please clarify how the advances in NLP and QCLA would impact patients rating free text as higher than rating scales (on page 12 of the discussion).

Our response: We have changed the first sentence in this paragraph to:

” The findings that participants rate free text response to be more precise in communicating mental health compared to rating scales is consistent with recent findings showing that QCLA has a high validity in measuring mental health.”

---

## [Decision Letter · Decision Letter 2]

16 Jan 2023

Precise Language Responses versus Easy Rating Scales - Comparing Respondents’ Views with Clinicians’ belief of the Respondent’s Views

PONE-D-22-11336R2

Dear Dr. Sikström,

We’re pleased to inform you that your manuscript has been judged scientifically suitable for publication and will be formally accepted for publication once it meets all outstanding technical requirements.

Kind regards,

Maki Sakamoto, Ph.D

Academic Editor

PLOS ONE

Additional Editor Comments (optional):

Reviewers' comments:

Reviewer's Responses to Questions

**Comments to the Author**

1. If the authors have adequately addressed your comments raised in a previous round of review and you feel that this manuscript is now acceptable for publication, you may indicate that here to bypass the “Comments to the Author” section, enter your conflict of interest statement in the “Confidential to Editor” section, and submit your "Accept" recommendation.

Reviewer #2: All comments have been addressed

2. Is the manuscript technically sound, and do the data support the conclusions?

Reviewer #2: Yes

3. Has the statistical analysis been performed appropriately and rigorously? 

Reviewer #2: Yes

4. Have the authors made all data underlying the findings in their manuscript fully available?

Reviewer #2: No

5. Is the manuscript presented in an intelligible fashion and written in standard English?

Reviewer #2: Yes

6. Review Comments to the Author

Reviewer #2: I would like to thank the authors for their diligent work on addressing the points made in the previous review! Some small edits remain (spelling and grammar mistakes, inconsistencies in the reference list, undifferentiated citations), but I believe that the paper is otherwise ready for publication.

7. PLOS authors have the option to publish the peer review history of their article (what does this mean?). If published, this will include your full peer review and any attached files.

Reviewer #2: No

---

## [Editor Report · Acceptance letter]

23 Jan 2023

PONE-D-22-11336R2 

Precise Language Responses versus Easy Rating Scales - Comparing Respondents’ Views with Clinicians’ belief of the Respondent’s Views 

Dear Dr. Sikström:

I'm pleased to inform you that your manuscript has been deemed suitable for publication in PLOS ONE. Congratulations! Your manuscript is now with our production department. 

Kind regards, 

on behalf of

Dr. Maki Sakamoto 

Academic Editor

PLOS ONE